# Local Differentially Private Release of Infinite Streams With Temporal Relevance

Anonymous

## ABSTRACT

The data stream generated by users on web applications is often collected using a local differential privacy (LDP) approach to ensure privacy. This approach offers rigorous theoretical guarantees and low computational overhead, albeit at the expense of data utility. Data utility encompasses both the value of individual data points and the temporal relevance that exists between them, but existing studies primarily focus on enhancing the former utility while neglecting the latter. Furthermore, the collected data often requires cleaning, and we have demonstrated through a case study that data stream lacking time relevance poses a significant risk to users' privacy during the cleaning process. In this paper, for the first time we present an online LDP publishing mechanism while preserving the inherent temporal relevance for the infinite stream, called the Sampling Period Perturbation Algorithm (SPPA). Specifically, we model the temporal relevance between data points as the Fourier interpolation function, resulting in a computational complexity reduction from $O(n^2)$ to $O(n \log n)$ when compared with the conventional Markov approach in the offline setting. To strike a better balance between privacy and utility, we add noise to the sampling period due to its minimal impact on sensitivity, which is analyzed by our novel concepts of $(\epsilon, \tau)$-temporal indistinguishability and $(\epsilon, w, \tau)$-event LDP. Through extensive experiments, SPPA exhibits superior performance in terms of both data utility and privacy preservation compared to the state-of-the-art baselines. In particular, when $\epsilon = 1$, compared with the state-of-the-art baseline, SPPA diminishes the MSE by up to 64.2%, and raises the event monitoring efficiency by up to 21.4%.

## CCS CONCEPTS

• **Security and privacy** → **Data anonymization and sanitization**; **Web application security**; • **Information systems** → **Data mining**.

## KEYWORDS

infinite streams, time series, temporal relevance, temporal privacy, local differential privacy

**ACM Reference Format:**
Anonymous. 2025. Local Differentially Private Release of Infinite Streams With Temporal Relevance. In *Proceedings of ACM Web Conference 2025 (WWW '25).* ACM, New York, NY, USA, 11 pages. https://doi.org/XXXXXXX.XXXXXXX

## 1 INTRODUCTION

In the realm of Web services, real-time data streams play a crucial role in delivering personalized experiences, targeted advertising, and other features such as real-time analytics, recommendation systems, and search engine optimization. However, with the increasing collection and analysis of vast amounts of personal data comes an elevated risk of privacy breaches. To address this concern, industry giants like Google and Apple commonly employ local differential privacy (LDP) techniques [5, 11, 15] to safeguard user data locally before uploading. Generally, a differentially private mechanism employs randomized algorithms, such as noise injection, to achieve data indistinguishability and thus provide privacy protection but reduce data utility. For example, noisy GPS trajectories can heavily affect downstream applications like classification [24] or traffic prediction [36]. Therefore, it is crucial to strike a moderate balance between privacy and utility [8, 22, 26, 28].

The utility of data streams lies not only in their value magnitude, but also in their implicit temporal relevance. For example, temperature changes throughout the day tend to rise and fall periodically over time, rather than occurring randomly. On one hand, temporal relevance is a key part of data dependencies, which are essential for identifying potentially faulty or inaccurate information within a given context [37]. Discovering these dependencies is a fundamental challenge in the data quality management pipeline, aiming to assist users to better profile the data and improve the data quality [9]. On the other hand, temporal relevance holds significant importance in data mining as well. For instance, analyzing user eye movements while browsing an e-commerce website can reveal implicit shopping behavioral patterns that, when better understood, can enable more precise targeted advertising [31].

However, current research predominantly focuses on enhancing the utility of statistics by changing the value of datapoints without considering their dependency [8, 19, 29, 35], resulting in a lack of temporal relevance in uploaded data streams. The drawback of these approaches is two-fold: For consumers relying on accurate information, preserving these dependencies is imperative; meanwhile for potential attackers who possess knowledge about such dependencies, there exists a possibility to compromise carefully designed privacy protection mechanisms. For example, with the primary intention of minimizing aggregation errors and enhancing utility, Ye *et al.* [35] propose an algorithm that shuffles datapoints in the stream in an LDP-compliant manner without changing actual values. However, in numerous scenarios (*e.g.,* taxi trajectory), the temporal relevance can be discovered similarly [9], making the reconstruction of the actual stream possible and thus resulting in a privacy catastrophe (see Section 2.4).

In this paper, we propose a novel real-time LDP algorithm for processing infinite streams without disturbing temporal relevance for the first time, called the Sampling Period Perturbation Algorithm (SPPA). Intuitively, time series data exhibiting temporal relevance can be viewed as a periodic sampling of a certain function $f$ with time serving as an independent variable. To publish an infinite stream satisfying LDP, the user can add noises to timestamps and resample the function $f$ using the perturbed timestamps before publishing them alongside the original real timestamps. In consideration of temporal relevance, SPPA introduces noises to the actual sampling period to ensure that all timestamps within a sliding window possess a uniform timestamp offset trend (either advance or lag), which guarantees that both the noisy timestamps and the resampled datapoints will not become disordered. To capture the differentially private characteristics of SPPA, we propose the novel notion of $(\epsilon, \tau)$-temporal indistinguishability, which measures the difficulty of locating the moment of an event. Furthermore, we introduce the notion of $(\epsilon, w, \tau)$-event LDP to quantify the capability of protecting any event sequence occurring within any window of $w$ timestamps in streaming settings.

In summary, we make contributions in this paper as follows:

- We shed light on the significance of preservation of temporal relevance in the LDP publishing mechanism for data streams, and demonstrate by a case study that these mechanisms are likely to pose a significant risk of privacy breach, which results from the data cleaning process using the additional knowledge of data dependencies.
- We are the first to propose an online LDP publishing mechanism while preserving the inherent temporal relevance for the infinite stream, which has been analyzed using Markov approach in the offline scenarios [7, 13, 38], but never in the online setting. We utilize the Fourier transform and interpolation techniques to preserve the sequential order of datapoints, reducing the computational complexity from $O(n^2)$ to $O(n \log n)$.
- To alleviate the impact of noise on data utility, we introduce novel concepts of $(\epsilon, \tau)$-temporal indistinguishability and $(\epsilon, w, \tau)$-event LDP, which offers a more precise description of privacy guarantee and helps to design LDP mechanisms with a better balance between privacy and utility.

## 2 BACKGROUND

We begin with a brief background on (local) differential privacy (DP/LDP) and their applications on infinite streams. Then we present a case study demonstrating that the LDP mechanism, when not preserving temporal relevance, poses potential privacy risks.

### 2.1 DP and LDP

The concept of differential privacy is initially proposed in the context of database query by Dwork [11], providing a rigorous guarantee that even for an adversary with sufficient prior knowledge, the original data cannot be inferred with high confidence from observing the output results. Under the LDP setting, an individual user needs to employ a randomized mechanism $\mathcal{M}$ to perturb the local data $v$ before publishing it to an untrusted server for downstream task. An LDP mechanism $\mathcal{M}$ must satisfy the following definition.

DEFINITION 1 (LOCAL DIFFERENTIAL PRIVACY, LDP [10]). *A mechanism $\mathcal{M} : \mathcal{D} \mapsto \mathcal{R}$, where $\mathcal{D}$ is the domain of all input $v$, satisfies $\epsilon$-local differential privacy ($\epsilon$-LDP), if and only if, for any pairs of $v, v' \in \mathcal{D}$, and any $O \subset \mathcal{R}$, it holds that*

$$\Pr(\mathcal{M}(v) \in O) \leqslant e^\epsilon \Pr(\mathcal{M}(v') \in O).$$

For any knowledgeable adversary, LDP guarantees that the user's original data cannot be inferred from the output of $\mathcal{M}$ with high confidence. Given any pair of inputs $v, v' \in \mathcal{D}$, the probability of "$v$ outputs $O$" is no more than $e^\epsilon$ times that of "$v'$ outputs $O$". The exponent $\epsilon$ is called privacy budget, and the smaller the $\epsilon$, the stronger the degree of privacy-preservation guaranteed by $\mathcal{M}$. The properties of DP are also inherited by LDP, with the two most commonly used ones being provided as follows.

THEOREM 1 (COMPOSITION [12]). *If $\mathcal{M}_i : \mathcal{D} \mapsto \mathcal{R}_i$ satisfies $\epsilon_i$-DP for any $i \in \{1, 2, \cdots, n\}$, then $\mathcal{M} = (\mathcal{M}_1, \mathcal{M}_2, \cdots, \mathcal{M}_n) : \mathcal{D} \mapsto \prod_{i=1}^{n} \mathcal{R}_i$ satisfies $\sum_{i=1}^{n} \epsilon_i$-DP.*

THEOREM 2 (POST-PROCESSING [12]). *Let $\mathcal{M} : \mathcal{D} \mapsto \mathcal{R}$ satisfy $\epsilon$-DP. Let $f : \mathcal{R} \mapsto \mathcal{R}'$ be an arbitrary randomized mapping. Then $f \circ \mathcal{M} : \mathcal{D} \mapsto \mathcal{R}'$ satisfies $\epsilon$-DP.*

Theorem 1 shows the additivity of privacy budgets, allowing us to combine multiple DP mechanisms, and Theorem 2 allows us to arbitrarily process data perturbed by LDP mechanisms without risk of increased privacy loss.

### 2.2 LDP on Infinite Streams

Originally, research on DP protects the presence or absence of an individual user (his/her entire data stream) for infinite streams, which is called user-level privacy [6, 14, 16, 17]. In contrast, the event-level privacy for finite streams protects the presence or absence of a single event in one user's data stream [12, 19, 20, 29]. Kellaris *et al.* [21] introduce a new notion of $w$-event DP for infinite event streams, to protect any $w$-neighboring prefixes over an infinite stream of "events" (i.e., data items generated by the users), and published periodically. These two privacy notions are extended to the local setting by Ren *et al.*, which strikes a good balance between user-level and event-level DP. Suppose a data owner has an infinite stream of time series $d = \{d_1, d_2, \cdots, d_n, \cdots\}$, with timestamps $t = \{t_1, t_2, \cdots, t_n, \cdots\}$. Let $S_n$ denote a stream prefix of $d$ up till time stamp $t_n$, $S_n = \{d_1, d_2, \cdots, d_n\}$. $S_{n,i}$ refers to the $i$-th element of $S_n$, i.e., $S_{n,i} = d_i$.

DEFINITION 2 ($w$-NEIGHBORING STREAM PREFIXES[26]). *Two stream prefixes $S_n, S'_n$ are defined to be $w$-neighboring, if for each $S_{n,i}, S_{n,j}, S'_{n,i}, S'_{n,j}$ with $i < j$, $S_{n,i} \neq S'_{n,i}$ and $S_{n,j} \neq S'_{n,j}$, it holds that $j - i + 1 \leqslant w$.*

The meaning of $w$ is twofold. First, $w$ is a sharp upper bound on the interval between the occurrence of two distinct elements; second, $w$ is also an upper bound on the count of distinct elements. Based on Definition 2, $w$-event LDP is defined as follows:

DEFINITION 3 ($w$-EVENT LDP [26]). *A mechanism $\mathcal{M} : \mathcal{S} \mapsto \mathcal{R}$, where $\mathcal{S}$ is the domain of all stream prefixes, satisfies $w$-event $\epsilon$-LDP (i.e., $w$-event LDP), if and only if, for any $n$, any pairs of $w$-neighboring stream prefixes $S_n, S'_n \in \mathcal{S}$, and any $O \subset \mathcal{R}$, it holds*

*that*

$$\Pr(\mathcal{M}(S_n) \in O) \leqslant e^{\epsilon} \Pr(\mathcal{M}(S'_n) \in O).$$

The definition of $w$-event LDP ensures that the adversary can not distinguish with high confidence between any two $w$-neighboring stream prefixes based on the output of mechanism $\mathcal{M}$. In practical implementation, the sliding window technique is commonly employed for achieving $w$-event LDP. This implies that even if a knowledgeable adversary possesses all the data in the stream except for a window of length $w$, after observing the perturbed window sequence satisfying $w$-event LDP, it would still not be able to reconstruct the original data with high confidence. Based on this concept, Kellaris *et al.* [21] design privacy budget allocation algorithms that have been always effectively applied in various DP circumstances [23]. Due to the good balance between privacy-preservation and application utility, numerous research follow the paradigm of $w$-event DP over infinite streams [8, 22, 26, 28, 30].

### 2.3 Time Series with Temporal Relevance

Time series data often have temporal relevance that can be exploited by adversaries, so that the privacy loss, *i.e.*, $\epsilon$, claimed by a DP algorithm is actually underestimated compared to the reality. Wu *et al.* [32] propose a privacy-preserving mechanism for trajectory correlation, which focuses on correlations between multiple users' trajectories. In contrast, our work investigates the correlations within an individual user's own data. Cao *et al.* [7] quantitatively characterize the privacy loss of temporal correlated time series in an offline setting, which cannot be applied directly in the online LDP scenarios. Zhang *et al.* [38] and Erdemir *et al.* [13] employ Markov models to formulate time series with temporal relevance. However, the strong temporal dependence among datapoints leads to significant growth in computational complexity, rendering it unsuitable for real-time publication.

Another approach to address the temporal relevance of the data involves utilizing the Fourier transform to convert the time domain relationship between datapoints into a frequency domain relationship, which can then be perturbed for privacy-preserving publication. Rastogi *et al.* [25] perturb the Discrete Fourier Transform of the query response. Ács *et al.* further improve this scheme by selecting the Fourier coefficients more effectively [2], and give an instantiation to publish the spatiotemporal density of the population of 989 different districts in Paris [1]. However, the approaches proposed in [1, 2, 25] rely on static data and are not well-suited for handling online infinite data streams.

### 2.4 A Case Study

The temporal correlation typically serves as public knowledge, which can lead to significant privacy breaches if exploited by adversaries. We will illustrate this argument with the following example.

Taxi trajectories[1] often exhibit a pronounced temporal correlation. We implement TSDDiscover [9] and find the following data dependency: $\forall t, \left( \Delta x_t \rightarrow^f_{(0,+\infty)} \Delta x_{t+1}, f(\Delta x_t) = \Delta x_{t+1} \right)$, where $\Delta x_t$ represents the displacement between the vehicle and its starting point. Intuitively, temporal relevance here equals to an order dependency that is a gradual increase in the displacement over time.

---

[1]https://www.kaggle.com/datasets/crailtap/taxi-trajectory

With the primary intention of minimizing aggregation errors and enhancing utility, Ye *et al.* [34] propose an algorithm that shuffles datapoints in the stream in an LDP-compliant manner without altering actual values. However, their approach, called ETM, neglects temporal correlations, resulting in a significant privacy loss when facing a knowledgeable adversary. To mimic the data cleaning process, we leverage the data dependency aforementioned and arrange the published data in an ascending order based on their distance from the starting point to the farthest, thereby significantly enhancing the resemblance to the true time series. This result is illustrated in Fig 1, which depicts the disparity between a perturbed (orange), cleaned (green), and the original (blue) sequence of a taxi trajectory. Hence, it is imperative to preserve temporal correlation rather than disrupting it when publishing perturbed sequences satisfying LDP.

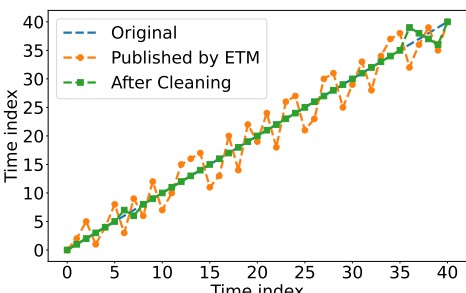

**Figure 1: The original sequence perturbed by ETM [35] can be readily inferred using temporal relevance. Due to the consistently increasing displacement over time, the data cleaning procedure can easily reconstruct the genuine user data.**

## 3 MODELS AND PROBLEM DEFINITION

In this section, we describe our data model and present our proposed notions of $\epsilon$-temporal indistinguishability and $(\epsilon, w, \tau)$-event DP. Then we formulate our problem definition.

### 3.1 Data Model

Consider a system consisting of distributed users and a central server. Every fixed period $T_s$, these users need to publish some data $d$ to the server for some kind of data mining task. As time evolves, each user produces an infinite stream of data $S = \{d_1, d_2, \cdots, d_n, \cdots\}$, with corresponding timestamps $t = \{t_1, t_2, \cdots, t_n, \cdots\}$. Note that the timestamps may not actually exist in reality or be uploaded to the server, but the server can still mark the data $d$ with its timestamp on receipt. Users consider these implicit time series to be sensitive and therefore prohibit the direct sharing of raw data with the server to preserve temporal privacy. For instance, a GPS service provider collects the location of individual users at each timestamp. Users aim to safeguard their location information, which can be accomplished through perturbing coordinates or timestamps, corresponding to the concepts of Value LDP (VLDP) and Temporal LDP (TLDP) as mentioned in [35].

## 3.2 Privacy Model

The goal of our privacy model is to protect the implicit timestamps corresponding to the data values. Let us begin with the simplest situation. Considering the perturbation of a single timestamp $t$, intuitively we want the probability distribution function of the perturbed version $t'$ to satisfy the following two properties:

(1) The expectation of $t'$ is equal to $t$: $\mathbb{E}t' = t$;
(2) The probability of $t'$ occurring decreases as the time interval from $t$ increases.

Following the Definition 1 of LDP, we define the notion of $\epsilon$-temporal indistinguishability.

DEFINITION 4 ($\epsilon$-TEMPORAL INDISTINGUISHABILITY). *A mechanism* $\mathcal{M} : \mathcal{T} \mapsto \mathcal{R}$ *satisfies $\epsilon$-temporal indistinguishability, if and only if, for any $O \subset \mathcal{R}$ and any pairs of inputs $t, t' \in \mathcal{T}$, it holds that*

$$\Pr(\mathcal{M}(t) \in O) \leqslant e^{\epsilon|t-t'|} \Pr(\mathcal{M}(t') \in O).$$

The notion of $\epsilon$-temporal indistinguishability satisfies the two properties intuitively summarized before. Note that the parameter $\epsilon$ here corresponds to the level of privacy at one unit of time, *i.e.*, it is the entire exponent $\epsilon|t - t'|$ that corresponds to the privacy budget. When $|t - t'|$ grows, the DP guarantee of the mechanism $\mathcal{M}$ degrades gracefully, akin to the concept of group privacy [12].

If we introduce the definition to call a pair of $t, t' \in \mathcal{T}$ such that $|t - t'| \leqslant \tau$ as $\tau$-*neighbor*, then $\epsilon$-temporal indistinguishability has the equivalent definition as follows.

DEFINITION 5 (($\epsilon, \tau$)-TEMPORAL INDISTINGUISHABILITY). *A mechanism* $\mathcal{M} : \mathcal{T} \mapsto \mathcal{R}$ *satisfies ($\epsilon, \tau$)-temporal indistinguishability, if and only if, for any $O \subset \mathcal{R}$ and any pairs of $\tau$-neighbors inputs $t, t' \in \mathcal{T}$ such that $|t - t'| \leqslant \tau$, it holds that*

$$\Pr(\mathcal{M}(t) \in O) \leqslant e^{\epsilon} \Pr(\mathcal{M}(t') \in O).$$

The notion of ($\epsilon, \tau$)-temporal indistinguishability gives the DP guarantee for all pairs of $\tau$-neighbors, which is similar to privacy definitions in [3, 23]. $\tau$ is the scope we provide privacy preservation, *i.e.*, data amongst $\tau$-neighbors are indistinguishable. $\epsilon$ in Definition 5 is numerically equivalent to $\epsilon\tau$ from the Definition 4.

We also require new notions of neighboring stream prefixes and event LDP, defined below by extending from Definition 2 and Definition 3.

DEFINITION 6 (($w, \tau$)-NEIGHBORING STREAM PREFIXES). *Two stream prefixes* $S_n = \{(d_1, t_1), \cdots, (d_n, t_n)\}$, $S'_n = \{(d'_1, t'_1), \cdots, (d'_n, t'_n)\}$ *are defined as ($w, \tau$)-neighboring, if*

(1) *the data of their elements are pairwise identical: for each $i \in [n]$, we have $d_i = d'_i$;*
(2) *the timestamps of their elements are $\tau$-neighboring: for each $i \in [n]$, we have $|t_i - t'_i| \leqslant \tau$ and*
(3) *all of the neighboring timestamps can fit in a window of time duration at most $w$ : for every $i < j$ with $t_i \neq t'_i$ and $t_j \neq t'_j$, it holds that $j - i + 1 \leqslant w$.*

DEFINITION 7 (($\epsilon, w, \tau$)-EVENT LDP). *A mechanism* $\mathcal{M} : \mathcal{S} \mapsto \mathcal{R}$, *where $\mathcal{S}$ is the domain of all stream prefixes, satisfies ($\epsilon, w, \tau$)-LDP, if and only if, for any $n$, any pairs of ($w, \tau$)-neighboring stream prefixes $S_n, S'_n \in \mathcal{S}$, and any $O \subset \mathcal{R}$, it holds that*

$$\Pr(\mathcal{M}(S_n) \in O) \leqslant e^{\epsilon} \Pr(\mathcal{M}(S'_n) \in O).$$

## 3.3 Problem Definition

Our goal is to design an ($\epsilon, w, \tau$)-event LDP solution that helps individual users to publish infinite data streams in real time. Simultaneously, we aim to minimize the mean error between the original and published data. In other words, the proposed algorithm should strike a good balance in the privacy-utility trade-off. To this end, we next present a framework satisfying ($\epsilon, w, \tau$)-event LDP achieving excellent utility.

## 4 PROPOSED TLDP MODEL

In this section, we elaborate on our proposed algorithm satisfying ($\epsilon, w, \tau$)-event DP and provide theoretical analyses on both privacy and utility.

## 4.1 Implementing TLDP via Laplace Mechanism

The Laplace mechanism is a classic technique for achieving $\epsilon$-DP by adding noise drawn from the Laplace distribution to the data before publication. Using the Laplace mechanism, we give the following theorem to achieve ($\epsilon, \tau$)-temporal indistinguishability.

THEOREM 3. *Given a timestamp $t$, the Laplace mechanism of outputting a perturbed timestamp $t'$ drawn from the Laplace distribution*

$$\text{Lap}\left(x; t, \frac{\tau}{\epsilon}\right) = \frac{\epsilon}{2\tau} \exp\left(-\frac{\epsilon|x - t|}{\tau}\right),$$

*satisfies ($\epsilon, \tau$)-temporal indistinguishability.*

The proof is presented in Appendix B. The noise variance of the Laplace distribution is determined by the privacy budget $\epsilon$ and the parameter of $\tau$-neighbors. The latter is close to the concept of sensitivity in the centralized DP setting [12], gauging the extent to which the data of a single entry can change the exposed information in the worst case. From this perspective, we can also refer to the parameter of $\tau$-neighbors as *time sensitivity*.

## 4.2 Single Window Perturbator

We start by designing an algorithm over a time series with $n$ timestamps to achieve ($\epsilon, \tau$)-temporal indistinguishability, called Single Window Perturbator (SWP). The time series data with temporal relevance can be perceived as periodic samples of a function, where time serves as the independent variable, *i.e.*, $d = f(t)$. The main idea of SPPA is to publish a new time series by resampling the function using the perturbed timestamps, *i.e.*, $d' = f(t')$. Since $t$ and $t'$ are $\epsilon$-temporal indistinguishable, from Theorem 2 of postprocessing it is guaranteed that $(d, t)$ and $(d', t)$ are also $\epsilon$-temporal indistinguishable.

Motivated by this insight, the technological process of SWP is illustrated in Algorithm 1 and Fig. 2. First, rather than perturbing each timestamp with an evenly allocated privacy budget, we choose to perturb the sampling period $T_s$ of the time series $S_n = \{d_0, d_1, \cdots, d_{n-1}\}$ (line 1). Thus, the timestamp sequence changes from $t = \{0T_s, 1T_s, \cdots, (n-1)T_s\}$ to $t' = 0T'_s, 1T'_s, \cdots, (n-1)T'_s$. This operation ensures a consistent temporal delay or advancement in the sampling process, corresponding to backward and forward releases as illustrated in Fig. 2(a) and Fig. 2(b). The order of the datapoints remains unchanged, but only undergoes shifts, thereby preserving the temporal relevance among the data. The privacy loss

---

**Algorithm 1:** SWP: Single Window Perturbator.

**Input** : True time series $S_n = \{d_0, d_1, \cdots, d_{n-1}\}$; True sampling period $T_s$; Time sensitivity $\tau$; Privacy budget $\epsilon$.

**Output**: Perturbed time series $S'_n = \{d'_1, \cdots, d'_{n-2}\}$.

Perturb the sampling period: $T'_s \sim \text{Lap}\left(x; T_s, \frac{\tau}{\epsilon}\right)$;

Compute the Fourier coefficients vector:
$\quad f \leftarrow \text{DFT}(d_0, d_1, \cdots, d_{n-1})$;

Compute the Fourier base frequency vector:
$\quad \omega \leftarrow \frac{2\pi}{nT_s}(0, 1, \cdots, n-1)^\top$;

**for** $i = 1$ **to** $n - 1$ **do**

$\quad$ Compute the perturbed data: $d'_i = \frac{1}{n} \sum_{k=0}^{n-1} f_k e^{j\,\omega_k i T'_s}$;

**end**

Return the perturbed time series: $S'_n = \{d'_1, \cdots, d'_{n-2}\}$.

---

will be analyzed later. Second, we use the discrete Fourier transform (DFT) to build the interpolation function $f(t; T_s)$ from time series data $S$ (lines 2 and 3). Finally, SWP resamples the Fourier interpolation function $f(t'; T_s)$ based on the perturbed sampling period $T'_s$ (line 4 and 5) and publishes the perturbed time series data $S'_n = \{d'_1, \cdots, d'_{n-2}\}$ (line 7). For each $S'_n$, we only publish the middle part of $(n-2)$ length, which is a combination of privacy and utility considerations. Specifically, the first datapoint is not protected due to $d'_0 = d_0$, while the periodic continuation nature of the DFT may lead to a significant deviation in $d'_{n-1}$ from its true value (as can be seen in Fig. 2), resulting in a reduction in utility. Therefore, we drop $d'_0$ and $d'_{n-1}$ from released stream. Again, we do not manipulate the timestamps in the published sequence explicitly.

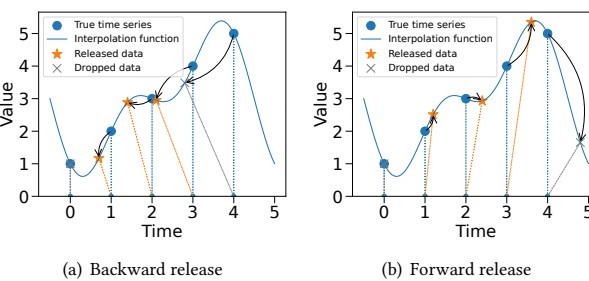

(a) Backward release $\qquad$ (b) Forward release

**Figure 2: SWP diagram. The SWP algorithm primarily comprises the following steps: (1) Perturbing the sampling period; (2) Calculating the Fourier interpolation function based on real datapoints (blue); (3) Determining the released sequence (orange) based on step (1) and step (2). Note that the first and last datapoints were dropped (grey) due to privacy and utility concerns, respectively.**

The reason why we choose DFT here to build the interpolation function is two-fold: Firstly, DFT can extrapolate function values beyond the time window through periodic extension when the number of datapoints is limited. Secondly, the utilization of fast

Fourier transform algorithms enables efficient implementation of DFT, rendering SWP suitable for the real-time release of LDP data.

### 4.3 Privacy Analysis

THEOREM 4. *Algorithm 1 satisfies $(\epsilon, \tau)$-temporal indistinguishability.*

The proof of Theorem 4 is presented in Appendix C. The role of $\tau$ is to quantitatively express the time extent guaranteed by the temporal DP, which was not previously included in the definition of TLDP[35], rendering their privacy guarantee vague. We will further discuss the significance and impact of $\tau$ in Sec. 5.2.

### 4.4 Utility Analysis

To assess the effectiveness of SWP, we quantify data value utility using the mean squared error (MSE) between real data value $d_i$ and published data value $d'_i$ as a conventional measure[25, 35]. The MSE is defined as follows: $\text{MSE}(d, d'_i) = \frac{1}{n} \sum_{i=0}^{n-1} (d_i - d'_i)^2$. The data value utility decreases as the MSE increases. Here we give the following result of utility analysis.

THEOREM 5. *The MSE of Algorithm 1 is*

$$\frac{1}{nw} \sum_{m=0}^{n-1} d_m^2 \sum_{i=1}^{n-2} \sum_{k=0}^{n-1} \frac{2}{1 + \left(\frac{\epsilon n T_s}{2ik\pi\tau}\right)^2}.$$

The proof is presented in Appendix D. It is evident from Theorem 5 that the utility of SWP improves with an increase in privacy budget $\epsilon$ and a decrease in time sensitivity $\tau$. The ratio $\tau/\epsilon$ characterizes the variance of the Laplacian noise, thus a higher variance leads to a reduction in MSE.

The influence of window size $w = n - 2$ is not readily apparent. The factor $\frac{1}{n} \sum_{m=0}^{n-1} d_m^2$ can be rewritten as the sum of mean and deviation of $\{d_0, d_1, \cdots, d_{n-1}\}$, so it can be regarded as independent with $w$. The remaining part can be approximated as $\frac{1}{w} \sum_{i=1}^{w} \sum_{k=1}^{w+1} \frac{2}{1 + \left(\frac{w+2}{ik}\right)^2}$, which increases with the window length $w$. We will validate this trend in our subsequent experiments.

### 4.5 Complexity Analysis

The computational complexity of the SWP in a single window is determined as follows. Let us consider having $n$ datapoints. In Algorithm 1, line 1 has a constant time complexity of $O(1)$. Lines 2 and 3 involve steps of the Discrete Fourier Transform, which have a time complexity of $O(n \log n)$. The inverse transformation from lines 4 to 6 also has a time complexity not exceeding $O(n \log n)$. Consequently, the overall computational complexity of SWP can be expressed as $O(n \log n)$. For comparison, the complexity of training the Hidden Markov Model (HMM) to characterize temporal relevance [33] is $O(n^2)$.

### 4.6 Sampling Period Perturbation Algorithm

Using a sliding window methodology, we deploy the SWP in the online setting to publish users' perturbed data satisfying $(\epsilon, w, \tau)$-event LDP over the infinite stream in real-time, called Sampling Period Perturbation Algorithm (SPPA). The pseudocode is presented in Appendix A.

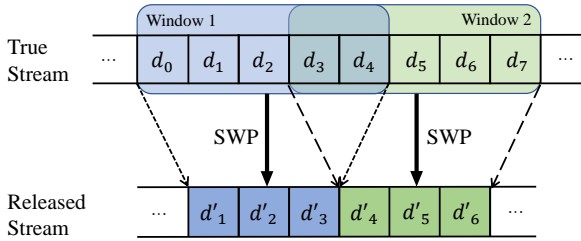

**Figure 3: SPPA diagram. SPPA employs sliding window technology to implement the SWP algorithm within each window independently. Despite the overlap between windows ($d_3, d_4$), the output sequence remains non-overlapping.**

The sliding window strategy of SPPA is illustrated in Fig. 3. In this example, the parameters $n_{perseg} = 5$ and $w = 3$ are used. Initially, the first window (in blue) consists of $\{d_0, d_1, d_2, d_3, d_4\}$ from the true stream. After applying SWP, only the perturbed data $\{d'_1, d'_2, d'_3\}$ with indices in the middle are published. Subsequently, the window moves forward by a step size of $w$, forming a second window (in green) containing $\{d_3, d_4, d_5, d_6, d_7\}$. Applying SWP to this second window results in publishing $\{d'_4, d'_5, d'_6\}$. The decision not to publish the first perturbed data generated within each window is based on its lack of privacy protection; specifically, it can be observed that $d_0 = d'_0$ in the first window and $d_3 = d'_3$ in the second window (note that $d'_3$ in the released stream originates from the first window). Additionally, for utility reasons, we choose not to publish the last perturbed data generated within each window due to its tendency to introduce large errors (see Fig. 2). It should be noted that there are overlaps with a fixed number of 2 between consecutive sliding windows on the true stream ($\{d_3, d_4\}$ in this example), which is a distinguishing feature compared to previous sliding windows strategies.

It is straightforward to show Algorithm 2 satisfies ($\epsilon, w, \tau$)-event LDP. By Theorem 4, the released sequence within each window satisfies ($\epsilon, \tau$)-temporal indistinguishability as defined in Definition 5. Moreover, the sliding window technique ensures that each released sequence is ($w, \tau$)-neighboring stream prefixes as defined in Definition 6. Combining these two properties leads to the conclusion that Algorithm 2 satisfies ($\epsilon, w, \tau$)-event LDP as defined in Definition 7.

## 5 PERFORMANCE EVALUATION

### 5.1 Experiment Settings

*5.1.1 Datasets.* To measure the actual performance of the SPPA, we conducted experiments on the following three real-world time series datasets: **Taxi**[2], **Eye**[27], and **Smartphone**[4].

**Taxi** dataset contains 1,710,670 different tracks generated by 442 taxis, publishing real-time GPS data at 15-second intervals. **Eye** dataset describes the eye movement data of 20 participants while reading different types of documents using VR headsets at a frequency of 30Hz. We utilize these two datasets to evaluate the data utility and event monitoring efficiency of different algorithms. **Smartphone** dataset records six activity tags (walking, sitting, etc.)

[2]https://www.kaggle.com/datasets/crailtap/taxi-trajectory

performed by 30 participants, as well as inertial sensor data from the smartphones they carried. The sensor data is published at a constant frequency of 50Hz in the form of 3-axis acceleration and 3-axis angular velocity. We utilize **Smartphone** dataset to evaluate the data utility and privacy-preserving capabilities of different algorithms, with the latter assessed through conducting subsequence clustering task [18] on the dataset.

*5.1.2 Baselines.* Following [35]'s naming of LDP mechanisms for different protection domains, we choose the following two algorithms to compare with SPPA.

- The Extended Threshold Mechanism (ETM) [35]. To the best of our knowledge, ETM is the only existing TLDP scheme that is directly applicable to our targeted setting.
- The Perturb-Group-Smooth algorithm (PeGaSus) [8]. It is a classic data-adaptive algorithm offering VLDP while maintaining improved accuracy for streams. We modified PeGaSus to make it suitable for local infinite streams.

### 5.2 Influence of $\tau$

One of the key parameters in our proposed SPPA algorithm, denoted as $\tau$, characterizes the level of privacy protection it offers by quantifying an adversary's ability to distinguish between noisy and real data. The theoretical analysis presented in Sec. 4.3 and Sec. 4.4 establishes that an increase in $\tau$ results in stronger privacy and lower utility (higher MSE). In this section, we will empirically verify this conclusion. The impact of window length $w$ and privacy budget $\epsilon$ on utility is deferred to the next section, as these parameters are also included in the other algorithms.

We conducted experiments on three datasets under 4 different settings: $(\epsilon, w) = (1, 4), (1, 8), (3, 4), (3, 8)$. Since the sampling frequencies of these datasets differ, we considered each sampling period as a benchmark and set different proportions: $\frac{\tau}{T_s} = 0.2, 0.4, 0.6, 0.8, 1.0, 1.2, 1.4, 1.6, 1.8, 2.0$. The experimental results are depicted in Fig. 4. Generally speaking, MSE exhibits an increasing trend with the increase in $\tau$, which validates our theoretical analysis.

### 5.3 Experimental Results

While smaller values of $\tau$ yield better utility, it also compromises the level of privacy protection. To strike a balance between the two factors, in subsequent experiments, we uniformly set $\tau = T_s$, where the sampling period $T_s$ is determined by each dataset.

*5.3.1 Data value utility.* We measure the data value utility by the MSE between real data stream $S$ and published data stream $S'$. A smaller MSE indicates better data value utility. Fig. 5 shows the performance of different algorithms in each dataset with different privacy budgets $\epsilon$. We fixed the window length to $w = 8$ and set $\epsilon = 0.5, 1, 1.5, 2, 2.5, 3$. The performance of different algorithms on all datasets exhibits a consistent trend, namely, an increase in errors with the rise of $\epsilon$, which reflects the trade-off between privacy and utility. A smaller privacy budget $\epsilon$ provides stronger privacy protection but leads to a larger MSE, indicating a reduction in data value utility. Moreover, both ETM and SPPA outperform PeGaSus in terms of data value utility due to their provision of TLDP which has less impact on the value of data than VLDP provided by PeGaSus. Furthermore, SPPA surpasses ETM with lower MSE across various

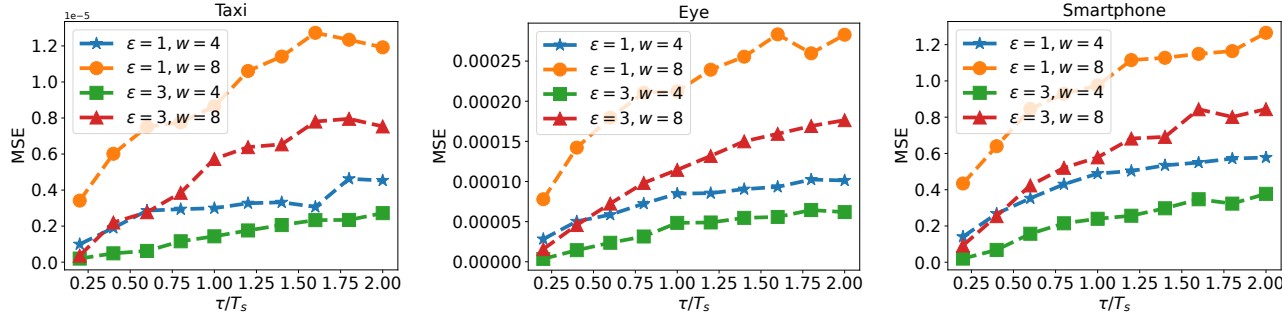

**Figure 4: Data utility with different $\tau$. Higher $\tau$ means a stronger promise of privacy, but also leads to lower utility.**

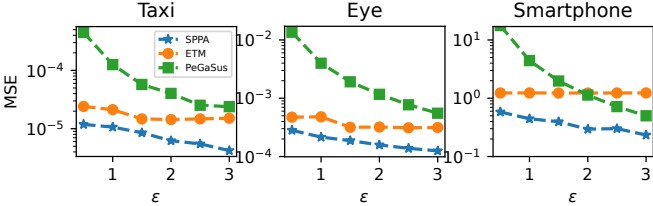

**Figure 5: Data utility with different $\epsilon$ ($w = 8$)**

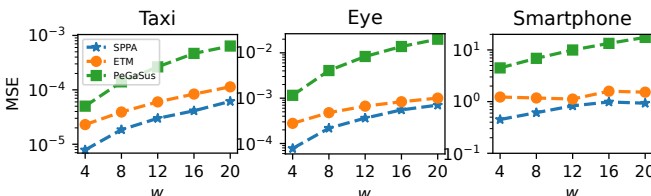

**Figure 6: Data utility with different $w$ ($\epsilon = 1$)**

privacy budgets and datasets because ETM tends to suffer from swapping distant data while SPPA smooths published data through Fourier interpolation of noisy time series, thereby reducing errors and improving utility.

Fig. 6 shows the performance of different algorithms measured by MSE in each dataset with fixed privacy budget $\epsilon = 1$ and different window length $w$. The value of $w$ should be carefully chosen to avoid excessive time delay. Specifically, the server needs to wait for $w$ time steps to fill a sliding window before implementing the LDP algorithm. Even without considering the computational cost and communication overhead of LDP, releasing data is still delayed by $(w - 1)$ time steps for the first data in the window. Therefore, we set $w = 4, 8, 12, 16$, and $20$. Generally speaking, as $w$ increases, the MSE of different algorithms also increases. This is because VLDP allocates less privacy budget per timestamp within each window while TLDP suffers from increased temporal sensitivity $\tau$, resulting in higher indistinguishability between distant timestamps. This trend reflects a trade-off between privacy preservation and utility enhancement since stronger indistinguishability leads to better privacy protection but worse utility performance. Notably, SPPA

consistently outperforms ETM across all values of $w$, demonstrating its superior performance.

*5.3.2 Event monitoring efficiency.* In this part, we fix $\epsilon = 1$ and $w = 8$. We implement the event monitoring to illustrate the algorithm's ability to preserve the temporal relevance of data stream. For **Taxi** dataset and **Eye** dataset, a common sense is that the velocity of the taxi or eye movement should not be excessively rapid. The data at the timestamp corresponds to the coordinate points in geography, so we first calculate the velocity between adjacent coordinate points, *i.e.*, $v_i = |d_{i+1} - d_i|/T_s$. Velocity above a given threshold $\delta$ is recognized as an anomalous event, which has different practical meanings in different situations, *e.g.*, speeding offense in **Taxi** dataset or saccading mode in **Eye** dataset. Fig. 7 provides an intuitional representation of utility by computing the relative velocity between each pair of coordinate points, arranged from left to right in order of original data, SPPA, ETM, and PeGaSus. The color blocks distributed along the diagonal approximately indicate the instantaneous speed of the vehicle per timestamp. It can be observed from Fig. 7(a) that the speed values for actual data were relatively low; however, after being subjected to ETM interference, a significant number of outliers emerged and instantaneous speeds reached a maximum of 500 km/h, which originates from exchanges of data order without considering temporal relevance. The PeGaSus algorithm and our SPPA algorithm lie somewhere in between, keeping the instantaneous speed within reasonable levels except for a few points. In Fig. 7(b) we also present the results from the **Eye** dataset. Although its temporal relevance may not be immediately apparent compared to the **Taxi** dataset, it still demonstrates a similar conclusion. Specifically, among the three algorithms, the SPPA noised sequence exhibits a location and number of dark areas that closely resemble those observed in real data.

We then quantitatively illustrate the event monitoring efficiency by plotting the receiver operating characteristic (ROC) curve. The area under the ROC curve (AUC), which is bounded by the diagonal line and the ROC curve, represents the algorithm's performance. A larger AUC indicates a better performance. We select different percentile velocities as the threshold $\delta$, which is calculated as $\delta_p \times (\max(v) - \min(v)) + \min(v)$ where $\delta_p$ ranges from 0 to 100%. Fig. 8 shows the ROC curve measured by the false positive rate (FPR) and true positive rate (TPR). In the **Taxi** dataset, SPPA achieves the highest AUC of 0.85, while ETM and PeGaSus obtain AUC of

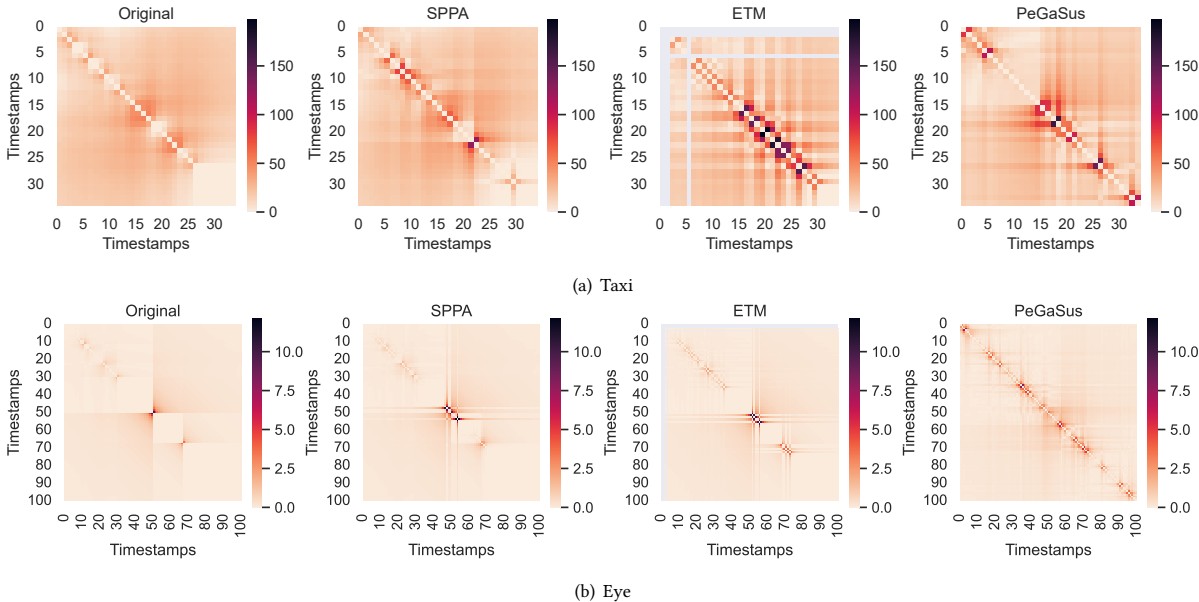

(a) Taxi

(b) Eye

**Figure 7: Heatmap for velocity between coordinates pairs in Taxi dataset and Eye dataset. In comparison to the original heatmap, a substantial disparity in speed at corresponding coordinates indicates a significant deviation in the published data. Each color block represents the average speed between the two timestamps, with darker colors indicating higher speeds.**

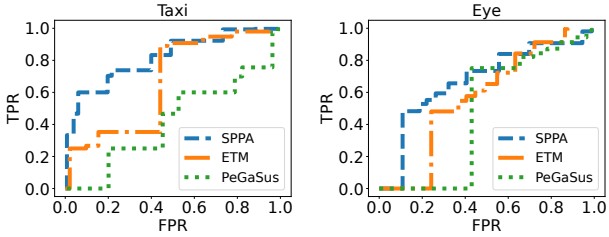

**Figure 8: ROC Curve for Event Monitoring ($\epsilon = 1, w = 8$)**

0.70 and 0.51, respectively. Similarly, in the **Eye** dataset, SPPA has an AUC of 0.74 compared to ETM's 0.65 and PeGaSus's 0.52. SPPA consistently demonstrates superior performance over both ETM and PeGaSus as evidenced by its larger AUC.

*5.3.3 Privacy preservation.* We adopt the TICC algorithm [18] to perform subsequence clustering task to examine the algorithm's ability of privacy preservation. With a focus on ensuring high levels of privacy protection, we set $\epsilon = 0.25, 0.5, 0.75, 1$, and $w = 8$. We use the F1 score to report the adversary's success in clustering of activity, as shown in Table 1. The lower the F1 score, the poorer the classification accuracy, indicating a weaker prediction of activity change and stronger privacy guarantee. When $\epsilon$ is fixed, the F1 score of true data is the highest, followed by data perturbed by PeGaSus and ETM, and SPPA has the lowest F1 score. Because TICC segments time series by identifying the time when the activity state changes, the lower the classification accuracy, the stronger the protection of temporal privacy. As the privacy budget decreases,

the privacy-preservation performance of all algorithms improves, with SPPA being the strongest.

**Table 1: F1 scores of predicting change point by TICC. A lower F1 score indicates a decreased classification accuracy and thus signifies the safeguarding of data.**

|  | $\epsilon = 0.25$ | $\epsilon = 0.5$ | $\epsilon = 0.75$ | $\epsilon = 1$ |
|---|---|---|---|---|
| Original |  | 0.813 |  |  |
| PeGaSus | 0.740 | 0.744 | 0.764 | 0.810 |
| ETM | 0.674 | 0.723 | 0.725 | 0.744 |
| SPPA | **0.582** | **0.634** | **0.640** | **0.706** |

## 6 CONCLUSION

Temporal relevance are frequently implied in infinite data streams, and the LDP framework, which disregards temporal relevance, exhibits significant security vulnerabilities. In this paper, we introduce the novel concept of $\epsilon$-temporal indistinguishability to quantify the differential privacy properties of a mechanism that protects temporal information. Additionally, we propose $(\epsilon, w, \tau)$-event LDP as a means to safeguard infinite streams and present the Sampling Period Perturbation Algorithm (SPPA) utilizing the sliding window technique, which achieves real-time $(\epsilon, w, \tau)$-event LDP for infinite streams with temporal relevance. We analyze SPPA's theoretical trade-off between privacy preservation and data utility and conduct numerous experiments to validate its performance. The experimental results demonstrate that our proposed algorithm effectively balances data utility and privacy preservation in the time domain.

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

## A  PSEUDOCODE OF SPPA

As shown in Algorithm 2, SPPA takes as input the true data stream $S = \{d_0, d_1, \cdots\}$, number of released datapoints per window $w$, true sampling period $T_s$, time sensitivity $\tau$, and privacy budget $\epsilon$. The length of a single sliding window is set to be $n_{\text{perseg}} = w + 2$ (line 1). Within each sliding window, SPPA executes the SWP algorithm to publish the perturbed time series data.

---

**Algorithm 2:** SPPA: Sampling Period Perturbation Algorithm.

| | |
|---|---|
| **Input** | : True data stream $S = \{d_0, d_1, \cdots\}$; Number of released datapoints per window $w$; True sampling period $T_s$; Time sensitivity $\tau$; Privacy budget $\epsilon$; |
| **Output** | : Perturbed data stream $S' = \{d'_0, d'_1, \cdots\}$. |
| **Initialize** | : Length of a single sliding window: $n_{\text{perseg}} = w + 2$; Window start position: $\texttt{start} = 0$; Window end position: $\texttt{end} = 0$; Window count: $i = 0$; Current window: $S_i = \{\}$. |

**while** $\texttt{end} < \texttt{len}(S)$ **do**
  add the next datapoint to the window:
   $S_i.\texttt{append}((d_{\text{end}}, t_{\text{end}}))$ ;
  $\texttt{end} = \texttt{end} + 1$;
  **if** $\texttt{end} - \texttt{start} == n_{\text{perseg}}$ **then**
    Execute perturbation algorithm:
     $S'_i = \text{SWP}(S_i, T_s, \tau, \epsilon, w)$;
    Release $S'_i$;
    Remove the first $w$ datapoints in $S_i$: $S_i = S_i[w:]$ ;
    $\texttt{start} = \texttt{start} + w$ ;
    $i = i + 1$ ;
  **end**
**end**

---

## B  PROOF OF THEOREM 3

PROOF. Let $t_1$ and $t_2$ are $\tau$-neighboring, *i.e.*, $|t_1 - t_2| \leqslant \tau$. For arbitrary timestamp $o$ drawn from the Laplace distribution defined in Theorem 3, we compare the probability density ratio

$$\frac{\Pr(o|t_1)}{\Pr(o|t_2)} = \frac{\exp\left(-\dfrac{\epsilon|t_1 - o|}{\tau}\right)}{\exp\left(-\dfrac{\epsilon|t_2 - o|}{\tau}\right)}$$

$$= \exp\left(\frac{\epsilon|t_2 - o|}{\tau} - \frac{\epsilon|t_1 - o|}{\tau}\right)$$

$$\leqslant \exp\left(\frac{\epsilon|t_2 - t_1|}{\tau}\right)$$

$$\leqslant \exp(\epsilon),$$

which satisfies $(\epsilon, \tau)$-temporal indistinguishability in Definition 5. The first inequality results from the triangle inequality, and the second inequality from the definition of $\tau$-neighbors. □

## C  PROOF OF THEOREM 4

PROOF. Given $\tau$ and $\epsilon$, we can achieve $(\epsilon, \tau)$-temporal indistinguishability for $T_s$ via line 1. For each periodic sampling point $d_i$, $i \in \{1, 2, \cdots, n - 1\}$, its corresponding timestamp is $t_i = t_0 + iT_s$. Since $T'_s$ obeys the Laplace distribution

$$\text{Lap}(x; T_s, \tau/\epsilon) = \epsilon/(2\tau)\exp(-\epsilon|T_s - x|/\tau),$$

then $t'_i = t_0 + iT'_s$ obeys the distribution as follows:

$$\text{Lap}\left(x; t_i, \frac{i\tau}{\epsilon}\right) = \frac{\epsilon}{2i\tau}\exp\left(-\frac{\epsilon|t_i - x|}{i\tau}\right),$$

which means Algorithm 1 satisfies $(\epsilon/i, \tau)$-temporal indistinguishability for $t_i$.

Note that the perturbation of timestamps is not independent, so the simultaneous release of $\{d'_1, \cdots, d'_{n-2}\}$ applies the Theorem 2 of post-processing, but not the Theorem 1 of composition. And since Algorithm 1 satisfies $(\epsilon/i, \tau)$-temporal indistinguishability for any $d'_i$ in published streams, it also satisfies $(\epsilon, \tau)$-temporal indistinguishability for all $d'_i$ in published streams by Definition 1. □

## D  PROOF OF THEOREM 5

PROOF. Using DFT we have

$$d_i = \frac{1}{n}\sum_{k=0}^{n-1} f_k e^{j\omega_k t_i}$$

and

$$d'_i = \frac{1}{n}\sum_{k=0}^{n-1} f_k e^{j\omega_k t'_i},$$

where $f_k$ is the $k$-th Fourier coefficients, j is the imaginary unit , $\omega_k = \dfrac{2k\pi}{nT_s}$ is the $k$-th circular frequency of DFT. So we have

$$\mathbb{E}\{\text{MSE}(S_n, S'_n)\}$$

$$= \mathbb{E}\left\{\frac{1}{w}\sum_{i=1}^{n-1}\left|\frac{1}{n}\sum_{k=0}^{n-1} f_k e^{j\omega_k t_i} - \frac{1}{n}\sum_{k=0}^{n-1} f_k e^{j\omega_k t'_i}\right|^2\right\}$$

$$= \frac{1}{n^2 w}\sum_{i=1}^{n-1}\mathbb{E}\left|\sum_{k=0}^{n-1} f_k e^{j\omega_k t_i}\left(1 - e^{j\omega_k(t'_i - t_i)}\right)\right|^2$$

$$\overset{(a)}{\leqslant} \frac{1}{n^2 w}\sum_{i=1}^{n-1}\sum_{k=0}^{n-1}\left|f_k e^{j\omega_k t_i}\right|^2 \mathbb{E}\sum_{k=0}^{n-1}\left|1 - e^{j\omega_k(t'_i - t_i)}\right|^2$$

$$= \frac{1}{n^2 w}\sum_{i=1}^{n-1}\sum_{k=0}^{n-1} f_k^2 \sum_{k=0}^{n-1}\mathbb{E}\left|1 - e^{j\omega_k(t'_i - t_i)}\right|^2$$

$$\overset{(b)}{=} \frac{1}{nw}\sum_{i=1}^{n-1}\sum_{m=0}^{n-1} d_m^2 \sum_{k=0}^{n-1}\mathbb{E}\left|1 - e^{j\omega_k(t'_i - t_i)}\right|^2,$$

where inequality (a) is from the Cauchy-Schwartz inequality and equality (b) is from Parseval's theorem of DFT. Now we calculate the expectation in the last line.

By wrapping up, the MSE can be bounded as in Theorem 5. □

$$
\mathbb{E}\left|1 - e^{\mathrm{j}\,\omega_k(t_i' - t_i)}\right|^2
$$

$$
=\mathbb{E}\left\{2 - 2\cos(\omega_k(t_i' - t_i))\right\}
$$

$$
=\int_{-\infty}^{\infty} \left(2 - 2\cos(\omega_k(t_i' - t_i))\right) \frac{\epsilon}{2\tau} \exp\left(-\frac{\epsilon\left|T_s' - T_s\right|}{\tau}\right) dT_s'
$$

$$
=2 - \frac{\epsilon}{\tau} \int_{-\infty}^{\infty} \cos(i\omega_k(T_s' - T_s)) \exp\left(-\frac{\epsilon\left|T_s - T_s'\right|}{\tau}\right) dT_s'
$$

$$
=2 - \frac{2\epsilon}{\tau} \int_{0}^{\infty} \cos(i\omega_k x) \exp\left(-\frac{\epsilon x}{\tau}\right) dx
$$

$$
=\frac{2}{1 + \left(\dfrac{\epsilon n T_s}{2ik\pi\tau}\right)^2}.
$$