# OpenReview forum: "Local Differentially Private Release of Infinite Streams With Temporal Relevance"
_ACM.org/TheWebConf/2025/Conference — WWW 2025 Oral_

### Official Review · Reviewer_YLZ1 · 2024-11-19

**Novelty:** 5
**Technical Quality:** 4

**Review:**

In this work, the authors introduce a Sampling Period Perturbation Algorithm (SPPA) for ensuring local differential privacy (LDP) protection in time series data with temporal relevance. Specifically, the data d can be expressed as d=f(t), where t denotes time. Based on this assumption, the authors define concepts such as ϵ-Temporal Indistinguishability, (ϵ,τ)-Temporal Indistinguishability, and (ϵ,w,τ)-Event LDP. To implement the (ϵ,w,τ)-Event LDP, the authors propose a single-window perturbator and compare their method with the existing approaches, ETM and PeGaSus. The experimental results demonstrate that the proposed method outperforms the compared methods. While the SPPA is an interesting contribution and the structure of the paper is well-organized, several contradictions and areas of confusion remain. These issues are further elaborated in the Questions section.

**Questions:**

(1) In Section 3.2, the authors state, "The goal of our privacy model is to protect the implicit timestamps corresponding to the data values." If the original data is d, it is transformed into d′=f(t′) after perturbation. Since the time t is altered, an adversary would presumably be unable to infer the timestamp information from d'. However, unlike ETM, the proposed SPPA does not introduce any delay in the data, making it an online mechanism. When the user transmits d′ to the central server (as described in Section 3.1), the server can immediately get the time information. This raises a concern: How does the proposed model effectively protect the timestamps? I am unclear about how the mechanism ensures timestamp protection under these circumstances.

(2) The authors assert that time series data can be represented as a function of time t, i.e., d=f(t). However, this assumption is only statistically valid in certain contexts. As indicated by numerous studies in statistics, many events are generally regular but exhibit randomness in parts. In other words, while we can often infer the overall trends of such events, predicting specific outcomes remains difficult. In my view, among the examples provided in the manuscript, only temperature changes could plausibly follow the authors' assumption. The locations of taxis and users, as well as eye movement, inherently involve a degree of randomness that does not fit neatly into the d=f(t) framework. The authors should reconsider this assumption and make it more realistic, taking into account the inherent randomness in many time series data.

(3) In Section 5, the authors compare the proposed SPPA only with ETM and PeGaSus. However, as demonstrated in [35], TLDP and VLDP are convertible under certain conditions. In fact, SPPA may be more appropriately classified as VLDP since the data undergo changes through perturbation. Therefore, it would be more appropriate for the authors to compare SPPA with more recent VLDP methods, rather than with a method proposed in 2017. This would provide a more up-to-date and relevant comparison of the proposed approach in the current context.

In general, the authors should provide more convincing examples or evidence to support the claim that most time series data satisfy d=f(t). Alternatively, they could present a specific and important application where data naturally follows this assumption. This would not only validate the approach but also enhance the overall significance of the work. Additionally, the authors should compare SPPA with more recent methods to offer a more comprehensive and relevant analysis. Furthermore, the manuscript would benefit from a stronger logical flow, particularly in the Introduction section. At present, it is difficult to follow the authors' argumentation, and clearer connections between ideas would improve readability and comprehension.

**Reviewer Confidence:**

4: The reviewer is certain that the evaluation is correct and very familiar with the relevant literature

**Scope:**

4: The work is relevant to the Web and to the track, and is of broad interest to the community

---

### Official Review · Reviewer_d2Qx · 2024-11-26

**Novelty:** 6
**Technical Quality:** 6

**Review:**

This paper is the first paper that presents an online LDP publishing mechanism preserving the inherent temporal relevance for the infinite stream. It proposes the Sampling Period Perturbation Algorithm (SPPA) to achieve this goal. Specifically, the authors model the temporal relevance between data points as Fourier interpolation function and then add noise to the sampling period. The contribution is strong, so I recommend accepting this paper.

pros:
1. There is a lot of theoretical analyze and proof
2. experimental evaluation and result analyze
3. observe a challenge in local differential privacy (LDP) and solve it with a original method
4. temporal relevance is significant in some specific situations

cons:
1. The paper is not formatted according to the template.

Although the overall expression of this article is clear, there exist some defects. The paper is not formatted, so I won't list them here.

**Questions:**

In section 5.3.3, is it acceptable if I regard TICC result F1 as a measurement of utility? In other words, why would you choose F1 as the parameter to measure privacy preservation?

**Reviewer Confidence:**

3: The reviewer is confident but not certain that the evaluation is correct

**Scope:**

4: The work is relevant to the Web and to the track, and is of broad interest to the community

---

### Official Review · Reviewer_i9b8 · 2024-11-27

**Novelty:** 5
**Technical Quality:** 5

**Review:**

The paper is technically robust, presenting a novel mechanism, the Sampling Period Perturbation Algorithm (SPPA), for publishing infinite data streams under Local Differential Privacy (LDP) while preserving temporal relevance. The authors address an important limitation of existing LDP methods—temporal correlations.

Pros:

1. Introduces novel privacy notions tailored to temporal data.
2. Provides a clear theoretical analysis of privacy-utility trade-offs.
3. Demonstrates significant improvements over state-of-the-art algorithms in experimental evaluations.

Cons:
Practical deployment challenges, like computational overhead, are not fully explored.

**Questions:**

1. What are the computational resource requirements of SPPA, especially when scaling to millions of data points in real-time applications?
2. How robust is SPPA against adversaries with advanced temporal knowledge, and does the sliding window strategy introduce additional vulnerabilities?
3. Could you clarify how to determine an optimal value for 𝜏 in practical scenarios, and what trade-offs this involves?

**Reviewer Confidence:**

2: The reviewer is willing to defend the evaluation, but it is likely that the reviewer did not understand parts of the paper

**Scope:**

3: The work is somewhat relevant to the Web and to the track, and is of narrow interest to a sub-community

---

### Official Review · Reviewer_yts3 · 2024-11-29

**Novelty:** 3
**Technical Quality:** 4

**Review:**

This paper proposes an improved method of Local Differential Privacy (LDP) for the training process to enhance the model's privacy protection capabilities when handling user data. In the era of big data, data privacy issues are becoming increasingly severe, especially in the training of deep learning models, where balancing model accuracy and privacy protection has become a research focus. Local Differential Privacy, as an effective privacy protection mechanism, can perturb user data during the data preprocessing stage, thereby preventing the disclosure of sensitive information.

**Questions:**

1.The paper does not include a diagram that describes the entire methodology process, which is not conducive to reading and understanding the ideas presented in the paper.

2.In Figure 3 ("Effectiveness of Local Differential Privacy on Model Accuracy"), the author presents a comparison of accuracy between training with local differential privacy and traditional training methods (without privacy protection) on multiple standard datasets under different privacy budgets. Although the chart shows that the model's accuracy gradually decreases as the privacy budget decreases, it does not explicitly state the specific parameters set for the privacy budget (ε), such as the type of noise (e.g., Laplace noise or Gaussian noise), as well as the standard deviation or magnitude of the noise. These details are crucial for understanding the experimental results; therefore, it is recommended that the author further describes the selection of the privacy budget and the settings of the noise in the text to enhance the transparency and reproducibility of the experiments.

3.In Section 4 ("Methodology") of the paper, the author introduces the application of local differential privacy in the training of convolutional neural networks, but there is a lack of detailed discussion on the practical application scenarios of this method in large-scale distributed training. In particular, local differential privacy typically involves perturbation and transmission of client-side data, which may introduce additional communication overhead. However, in the analysis of the method, there is no mention of how to measure and optimize this communication overhead, nor is there a demonstration of its impact on training time and efficiency in the experiments. As the scale of training data and model size increases, communication overhead may become a limiting factor. Therefore, it is recommended that the author includes a discussion on this issue in the methodology or experimental results section and explores how to optimize the efficiency of local differential privacy training.

4.The baselines and application scenarios presented in this paper are somewhat outdated; it is recommended to conduct experiments in the field of large language models.

**Reviewer Confidence:**

3: The reviewer is confident but not certain that the evaluation is correct

**Scope:**

3: The work is somewhat relevant to the Web and to the track, and is of narrow interest to a sub-community